# An Energy-Efficient T-Based Routing Topology for Target Tracking in Battery Operated Mobile Wireless Sensor Networks

**DOI:** 10.3390/s23042162

**Published:** 2023-02-14

**Authors:** K. Kalaivanan, G. Idayachandran, P. Vetrivelan, A. Henridass, V. Bhanumathi, Elizabeth Chang, P. Sam Methuselah

**Affiliations:** 1School of Electronics Engineering, Vellore Institute of Technology, Chennai 600127, Tamilnadu, India; 2Department of Electronics and Communication Engineering, Anna University Regional Campus Coimbatore, Coimbatore 641046, Tamilnadu, India; 3School of ICT, Griffith University, Gold Coast, QLD 4222, Australia; 4School of Computer Science and Engineering, Vellore Institute of Technology, Chennai 600127, Tamilnadu, India

**Keywords:** camera, energy efficiency, land mine, T-based routing topology, target tracking, wireless sensor networks

## Abstract

Real-time smart applications are now possible because to developments in communication and sensor technology. Wireless sensor networks (WSNs) are used to collect data from specific disaster sites, such as fire events, gas leaks, land mines, earthquake, landslides, etc., where it is necessary to know the exact location of the detected information to safely rescue the people. For instance, the detection and disposal of explosive materials is a difficult task because land mines consistently threaten human life. Here, the T-based Routing Topology (TRT) is suggested to gather data from sensors (metal detectors, Ground Penetrating Radars (GPR), Infra-Red sensors, etc.), Global Positioning System (GPS), and cameras in land mine-affected areas. Buried explosive materials can be found and located with high accuracy. Additionally, it will be simpler to eliminate bombs and reduce threats to humans. The efficiency of the suggested data collection method is evaluated using Network Simulator-2 (NS-2). Also, the proposed T-based routing topology requires a minimal number of nodes to cover the entire searching area and establish effective communication. In contrast, the number of nodes participating in the sensing area grows, as the depth of the tree increases in the existing tree topology-based data gathering. And for cluster topology, the number of nodes deployment depends on the transmission range of the sensor nodes.

## 1. Introduction

Recent developments in sensor and communication technologies are playing a vital role in many smart applications due to their adaptability, low cost, quick expansion, and availability [1]. Wireless Sensor Networks (WSNs) are frequently used to gather data from the target region and transmit it to the sink. The capability of a sensor node to self-configure with its neighbours and build the network without human intervention is one of its key characteristics [2,3]. Sensor nodes are placed in the intended region to collect the physical parameters like pressure, temperature, vibration, light, humidity, etc. Sensor technology is essential to many real-time applications, including smart farming, smart aquaculture, earthquake monitoring, gas leak detection, nuclear radiation detection, landslide detection, tsunami detection, and so on [4,5,6,7].

One of the dangers to human life around the world is the use of land mine bombs. A land mine bomb is an explosive device used in battle to destroy enemy targets and vehicles that is buried in or on the ground [8]. The Royal Demolition Explosive (RDX), Trinitrotoluene (TNT), Tetryl, High Melting Explosive (HMX), acetone peroxide, a metal or plastic container, and a pressure sensor are the main components of the land mine. It is divided into two categories: (i) antipersonnel land mines and (ii) anti-vehicle land mines. The antipersonnel land mine, often called VS 5.0, PMD 6, VAL 69, or PMA3, is a small, explosive device that bursts into flames when stepped on or disturbed. It is used by the military to harm the enemy. VS 1.6, TMA 4, M15, and M19 are a few examples of anti-vehicle land mines that are buried beneath the earth of the roadway to target the trucks and cars of the opposing side [9,10]. Landmine is electrically triggered and incurs significant monetary and personal losses when the heavy vehicle passes over them. The unexplored land mines remain a threat, even after the war is over.

Numerous mine awareness initiatives were established to inform the public and promote safe conduct among residing communities. According to the United Nations (UN), it would take approximately 1100 years to entirely remove all hidden land mines from the earth. According to the land mine monitor database, more than 12,000 casualties from anti-personnel and anti-vehicle land mines were reported between 1999 and 2017. Table 1 shows the list of countries that are most affected by the land mine and the total number of casualties reported [8].

The present routing algorithms are closely related to static WSNs which deploy numerous sensor nodes to expand coverage and improve data accuracy. When using the present random node deployment and routing technique, it is very difficult to pinpoint the exact position of events in the hazardous region, such as the detection of land mine bombs, gas leaks, and radiation emitting sources. Furthermore, the mobile WSN’s random node placement is not sufficient to provide full coverage for the land mine search. Furthermore, it does not guarantee that the location being tracked will be secure or qualified as a safe zone. According to the UN, finding the exact location of the buried land mine is still required in order to dispose of the explosive safely. The land mine is typically located using a metal detector, a dog, a rat, and a person. However, if the land mine is not located precisely, it can cause severe damage, death and financial loss.

In order to address these problems, it is planned to employ a completely automated technology to locate and destroy landmines without endangering people. An energy-efficient T-based Routing Topology (TRT) is proposed, in which the robotic vehicle incorporates the sensor nodes and Global Positioning System (GPS) to find and lock the precise location of land mines.

The major goal of this paper is to create networks in the targeted area and to gather information about the buried device, including its precise location and picture. This technique use a metal detector to locate the landmine, and a GPS map is utilized to pinpoint exactly where it is buried. The operator can safely detonate the land mine by using the camera to record the location of locked land mines. The following are the main contributions of the suggested methods:Dividing the network area into strips to minimize the number of deployed sensor nodes and achieve effective tracking of the objects.Measuring the moving speed of sensor node based on the sensing range and length of the detector’s stick.Introducing T-based Routing Topology (TRT) to ensure the congestion free data collection.Analyzing the performance of the proposed data routing protocol with respect to the Packet Delivery Ratio (PDR), Average End-to-End Delay (AEED), Average Energy Consumption (AEC) and control overhead.

The remainder of the paper is organized as follows: Section 2 provides related works. Problem statement is given in Section 3. Section 4 presents the proposed protocol design. Section 5 gives mathematical analysis of the proposed design. Section 6 provides the simulation results and discussion. Finally, the conclusion is presented in Section 7.

## 2. Related Works

There have been many strategies used recently to locate the buried land mine. The dog’s primary training involves finding the RDX-exploded substance in the target area. it has a 60-centimeter detecting range for objects buried in the ground. The rats were taught to scratch a hidden area on their foot to indicate the presence of land mines by rewarding them with food every time they did so. A catastrophic loss could occur if buried weapons are not precisely located due to poorly trained dogs and rats. Additionally, various types of explosive material in the affected area were found using bees, bacteria, and plants [9,10,11].

Radio waves were used by the ground penetrating radar to examine the characteristics of a return signal from the explosive components. Infrared detection picks up the electromagnetic variation in the buried explosive surface due to the signal’s reflection and emission. The temperature variation close to the explosive’s buried surface was recorded by the thermal-based detector. Due to the fact that different materials exhibit different types of signal reflection, the ultrasound-based detection generates a variety of acoustic properties depending on the materials [9,10]. One of the most significant challenges in real-time data collection is determining the exact position of an event’s occurrence, hence many applications employ the camera and GPS to determine the location. However, the use of GPS and image processing techniques increases computing complexity, memory, and bandwidth, which are directly related to the energy consumption in WSNs. A number of camera-based technologies have been developed with image compression methods in order to improve network longevity for energy-constrained WSNs. These techniques increase the rate of packet delivery by reducing the amount of bits in the image signal. The optimal zonal 2 × 2 BinnDCT was proposed for WSNs in [12], in which a shift and add function is used to extract the coefficients of an 8 × 8 image block. A Discrete Wavelet Transform (DWT) based distributed image compression technique was proposed in [13] to share the processing tasks and aimed to reduce the computational complexity and energy consumption. A distributed picture compression method based on the JPEG 2000 standard was presented in [14], in which parallel DWT and tiling techniques were utilized to exchange information and distribute the processing workload between the different nodes. The Lapped Bi-orthogonal Transform (LBT) based image compression was introduced in [15] which utilized a clustering technology to distribute the processing load across the nodes and lower the large amount of energy consumption. Numerous routing schemes were proposed to gather the information and transmit it to the Base Station (BS) [16,17,18]. For instance, the Unmanned Aerial Vehicle (UAV) or Unmanned Surface Vehicle (USV) was employed to monitor nuclear radiation in a specific area and identify regions of radiation that were highly concentrated.

The Internet of Things (IoT) has been designed with the standard Transmission Control Protocol (TCP) to offer secure data connections without traffic congestion. However, it could not be used directly to WSNs due to the restrictions imposed by sensor nodes. For the analysis of mobile ad-hoc networks, a probabilistic, energy-aware broadcast calculus is proposed in [19,20] to enhance topology control and lessen interference. The formulation and analysis of the connectivity and energy consumption is a challenging research topic in MANET. This can lead to design an automatic tool, as described in [21]. Also, a design level formal model is described in [22,23] to avoid the collisions and interference. An appropriate routing and topology structure is required to prevent the collision, enhance the packet delivery rate and throughput.

In traditional WSNs, the sensor nodes were manually or randomly placed in remote areas to sense the necessary events and send the data directly to the BS. The main disadvantages of direct communication are high power consumption and short network lifetime [24]. To overcome this issue, the multi-hop routing approach was developed to extend the network lifetime. However, it also addresses the problem of temporal synchronization due to the lack of centralized coordination. As a result, the collisions and interference may occur if numerous sensor nodes attempt to send their sensed data to the sink at the same time [25]. Additionally, during the setup phase, each node makes an effort to create a route in a proactive or on-demand way by exchanging a large number of control packets, resulting in high energy and memory consumption [17].

One of the most popular routing mechanisms utilized in WSNs was Low energy Adaptive Clustering Hierarchy (LEACH) [24], in which the clustering mechanism was introduced to improve the energy efficiency and scalability of networks. The LEACH protocol fails to distribute the cluster head (CH) uniformly as a result of the CHs’ random rotation, which leads in link failure and isolated node issues. To overcome this issue, the waiting time based CH selection was introduced in [26] which ensures load balance and even energy consumption, thereby extending the network lifetime. Mobility Based Clustering (MBC) was presented in [27] to select CHs based on residual energy and speed as similar to that of LEACH. In MBC, inter-cluster communication is carried out using the CSMA/CA through the defined tree topology; thus it encounters a large delay. Efficient Distributed Clustering and Gradient based Routing Protocol (EDCGRP) was proposed in [28] to collect the data from sensor nodes periodically. Also, the CHs were selected based on the node degree, residual energy and received signal strength. Energy aware clustering and efficient CH selection was proposed in [29], in which the CHs were selected based on the distance to BS, distance to local neighbours, and remaining energy. The residual energy prediction with fuzzy logic based clustering method was proposed in [30] to uniformly distribute the work load of the sensor node and improve the network lifetime. The optimal selection of the CHs in the grid was proposed in [31], in which the sensing area is divided into cluster grids and the CHs selection is based on their distance from the BS and remaining energy. The moving speed, residual energy, and pass time were proposed as the input parameters for an improved LEACH protocol that uses fuzzy logic to increase the packet delivery ratio [32]. Several data aggregation strategies have been proposed for cluster-based WSNs to eliminate redundant packets. The choice of data aggregation techniques is determined by the energy requirements of the application. In WSN, the cluster based data aggregation eliminates redundancy in the sensed data, in which the cluster head collects the data from cluster members and sends the aggregated data to the base station [33]. As a result, it avoids sending of the entire sensed data and conserves significant amount of battery energy [34]. However, the majority of the currently used protocols do not take into account the sensor’s position information, making it difficult for them to function well in real-time strategies because the sensed data is useless without a valid location (e.g., rescue applications). Data communication mode is related to the three different types of WSN applications, such as event-driven, periodic, and on-demand reporting. In the event-driven mode, the sensor node sends data to the base station whenever a specified event (for example, a gas leak) takes place in the sensing region. In a periodic drive, the sensor nodes send data to the BS on a regular basis at specified intervals (for example, environmental monitoring). In the demand-driven mode, the BS transmits commands to the specified sensor nodes, indicating that they ready receive data (for example, stock verification at storage room) [26]. The data aggregation method is more suitable in periodic driven applications and it is ineffective in event driven based application like landmine detection.

The clustering algorithm and Travelling Salesman Problem (TSP) based routing was presented in [35] to establish the path between sensor nodes and BS through UAV. In [36], LoRa communication technology was utilized to establish the communication between the UAV and on-ground sensors and collect the data from the agriculture field periodically. Policy-iteration based path planning algorithm was applied in routing formation which avoids the inter-unmanned surface vehicle collision [37]. A bi-level hybridization-based metaheuristic algorithm was deployed in [38] to find the tour planning and avoid the large blind searches in complex space. Neuro fuzzy based cluster formation and metaheuristic route formation was presented in [39], in which the remaining energy, distance to UAV and degree of UAV to cluster are considered in path planning. Voronoi diagram based path planning algorithm was used to linger the UAV’s location with minimal computational effort to collect the data from the sensor node [40]. The water quality detection and sampling was performed using unmanned surface vehicle in which the particle swam optimization was used to find the feasible data collection path of seawolf USV [41]. In [42], the IoT-UAV grouping was executed using k-means clustering algorithm, in which the path contains optimal value of the completion time and energy consumption. The query based data collection scheme was presented in [43] to select the particular node based on the query and then send the UAV for collecting the sensed information. The comparison of existing methodology with the proposed TRT is given in Table 2.

From the literature survey, it can be summarized as follows:Numerous routing protocols have been evolved in the past years to collect the data from the targeted area. However, the existing protocols are not suitable for all the applications in real time. For example, in tree topology based data collection, the number of nodes involved in each level is $2^{n}$. The number of nodes participating in the sensing area grows $\left(\sum_{i=0}^{n}2^{i}\right)$ as the depth of the tree increases in the existing tree topology-based data gathering. And for cluster topology, the number of node deployment depends on the transmission range of the sensor nodes.The majority of research on WSNs focuses on periodic data collection, in which all sensor node nodes send data to the base station at regular intervals.Most of the protocols utilise a large number of sensor nodes to collect data or detect an event in a specific area.The majority of protocols are designed for a standard application and are intended to collect data from the sensing area and transmit it to the base station in a multihop fashion.

In this paper, the T-based routing algorithm is specifically designed for collecting the data in landmine detection which utilizes a minimal number of nodes to cover large area and establish an effective communication using strip based network. It also extends the communication between the sensor node and BS by deploying the router nodes.

## 3. Problem Statement

Traditional methods use a metal detector, dog, rat, and human to locate the land mine, but failing to pinpoint its exact location result in catastrophic harm, including damage, death, and financial loss. To solve these issues, it is planned to deploy a fully automated system for collecting the data efficiently without affecting the human lives. The unmanned surface vehicle contains the sensor nodes to locate and lock the exact position of the land mines using the Global Positioning System (GPS). The camera, GPR, electro-magnetic induction sensor and GPS are used to capture the location of locked land mines that enables the operator to safely detonate the land mine. Still what is missing..?

Poor data communication between the sensing node and base station is being caused by a lack of network coverage.The data routing and location identification of landmine in real time is limited.Still, it has wrong location marking, low operating speed and effectiveness.

## 4. Proposed Protocol Design

This work introduces an energy-efficient T-based routing topology that searches and locates land mines precisely utilizing camera-based WSNs and GPS.Additionally, the strip-based network partitioning is implemented to facilitate the detection of land mines and to improve network coverage with a minimal deployment of nodes. The user can also locate a precise location and minimize the hazards to human life by using the camera image. The general diagram of proposed model is shown in Figure 1.

### 4.1. Network Assumptions

The major assumptions of the networks are (i) All sensor nodes are aware of its exact location in the targeted sensing region. (ii) All sensor nodes are identical in terms of mobility, memory, bandwidth, battery life, and compression technique. (iii) The transmission range of the sensor nodes can be adjusted depending on the distance between them (iv) The base station is located at the sensing zone border and the entire sensor node can interact directly with BS.

### 4.2. Network Model

The base station is positioned at one end of the network boundary. All sensor node moves in the same direction at a constant speed. The links between the nodes in the network model are represented by an undirected connectivity graph *G* and the vertices. Additionally, the node’s speed, direction of motion, transmission range, and location are taken into account as representations of the sensor nodes in the sensing region using Cartesian coordinates. The robotic vehicle with sensor nodes is shown in Figure 2.

The targeted sensing area is divided into the strip based on the length of sensor/detector holding pole $\left(d_{l}\right)$ and height from the ground $\left(d_{h}\right)$. By using the pythagoras theorem, the distance $\left(d_{c}\right)$ is calculated as given in Equation (Equation 1).
(1)$$d_{c}=\sqrt{{\left(d_{l}\right)}^{2}-{\left(d_{h}\right)}^{2}}$$

In this method, the rotating part can rotate the sensor holding pole between $0^{\circ }$ and $180^{\circ }$, therefore $d_{c}$ is considered as radius. And its diameter is used to divide the sensing area into the strips and the number of required mobile sensor nodes $\left(N_{d}\right)$ is calculated as
(2)$$N_{d}=\frac{M}{2d_{c}}$$
where *M* is the breadth or length of the network area.

The number of router nodes required $\left(N_{d}\right)$ is calculated as
(3)$$N_{r}=\frac{M}{t_{r}}$$
where $t_{r}$ is the transmission range of sensor node.

The total number of nodes (both mobile nodes and router nodes) required to cover the searching area is calculated as
(4)$$N_{T}=N_{d}+N_{r}$$

The network region is splitting into the strips and each strip is covered by a single Mobile Node (MN). From the deployment, each node knows its direct hop neighbors. The mobile node in the central strip is called as Coordinator Node (CN) that performs two operations: (i) coordinating the operation of nodes (ii) deploying the router node. The main role of coordinator is to deploy the router nodes based on the transmission range $t_{r}$, in which CN tracks the traveled distance, and it drops the router nodes once its traveled distance reaches the transmission range. By this way, the CN expands the network searching area and establishes the communication with the BS. Routers are static nodes which are used to transfer the data to the BS in a multi-hop manner. Also, the coordinator collects the data from other mobile nodes which means that it creates a horizontal path between the CN and other MNs. Then, the coordinator node transmits data vertically to the BS through router nodes. The distance to be traveled by the robotic vehicle at each round is measured depending on the sensing range $S_{r}$.

### 4.3. Energy Consumption Model

The communication energy consumption model utilized in the proposed model is similar to that of [24]. The energy consumption for receiving the data packets $\left(e_{r}\right)$ is measured based on the size of data packet, as given in Equation (Equation 5).
(5)$$e_{r}=k\times e_{elec}$$
where $e_{elec}$ is the energy dissipation due to the electronic circuits such as encoder, decoder, multiplexer, demultiplexer, and filter, *k* is the length of packets. The energy consumption for transmitting the packets $\left(e_{t}\right)$ is based on the size of packet and distance between the transmitter and receiver *d*, as given in Equation (Equation 6).
(6)$$e_{t}=k\times e_{elec}+k\times \epsilon \times d^{\alpha }$$
where ϵ is the transmitter amplifier energy based on the propagation model (free space $\epsilon_{fs}$ or multi-path $\epsilon_{mp}$ fading), α is amplifying factor, for free space model α=2, when $d\leq d_{0}$ and for multi-path fading model, α=4, when $d>d_{0}$, where $d_{0}=\sqrt{\frac{\epsilon_{fs}}{\epsilon_{mp}}}$.

### 4.4. T-Based Routing Topology

In the proposed method, the metal detector is attached to the pole with GPS, and camera is mounted on the vehicle head. The metal detector pole is rotated from $0^{\circ }$ to $180^{\circ }$ in the moving direction. If the metal detector detects the land mine, then the moving pole is immediately locked the particular place. The locked location is identified with the help of GPS information and the camera is used to take a photo of the marked location. Finally, the USV sends the GPS information and photo the BS through the router node. Using this information, the trained person easily reaches the landmine buried location and removes safely.

This article mainly focuses to cover the entire searching area with minimal numbers of sensor deployed USVs. For this, the sensing area is divided into strips based on the sensing range of metal detector and pole length. The number of strips in the sensing area is derived based on the Equation (Equation 2) and each strip is covered by a single USV which means that the number of USVs is equal to the number of strips. The CN moves vertically alongside its neighbouring deployed nodes and drops the router nodes based on the transmission range, so that it can easily establish the multi-hop T-based routing topology. Thus, it ensures the connectivity between the mobile nodes and BS, as shown in Figure 3.

The proposed mechanism makes use of event-based data collection mode by transmitting the Path Reservation Message (PRM), in which the source node reserves the routing to the CN when it detects the events. Only the direct neighbours of each node are aware of information like neighbours ID and distance. Initially, the source node transmits PRM using its default transmission range, which includes the source node ID and hop count. The nodes located in the source node transmission range open the PRM message and determine whether it is from its immediate neighbour or not. The source node identifies that the next hop node has been successfully reserved when it receives the acknowledgment (ACK) from its direct-hop neighbour before the timeout. If the source node does not receive the acknowledgment from its immediate neighbour, then it retransmits the PRM with incremented hop count value (HC = 2). It means that the source node receives the ACK from its second-hop neighbours and other nodes in the tr of source nodes will discard this PRM. If the source node does not receive the ACK before the maximum attempt, then it will expand the transmission range by Xtr, where (X = 1, 2, 3…). By this manner, the intermediate nodes reserve the routing to BS. Additionally, the sensor nodes can adjust the power of their data transmission based on the received signal strength of PRM and ACK. The process of PRM-ACK based routing is described in Algorithm 1.
**Algorithm 1:** PRM-ACK based route formation.
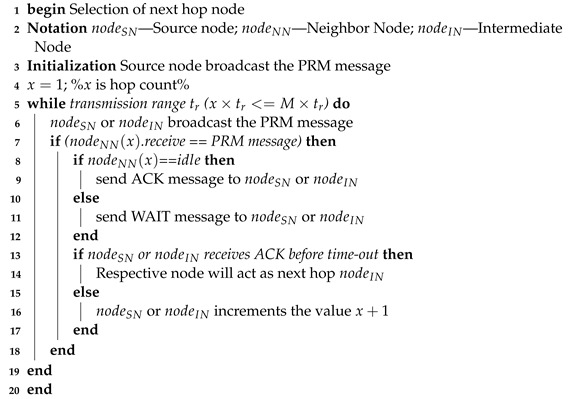


As shown in Figure 4, node *B* is currently sending its data to the CN, at the same time node *C* wants to send its data and node *C* sends PRM to node *B*. But, node *B* sends back the wait message instead of ACK which means that node *B* either currently sends its own data or processes the PRM message of other node. It therefore avoids the collision and interference. Another case, node *A* detects the event during the process of node $B^{\prime }s$ PRM and then node *A* knows itself to wait for completing the node $B^{\prime }s$ data transmission.

It is noted that from Figure 5, node *B* sends PRM to reserve the path when it wants to send the data. And node *A* doesn’t send ACK or wait message before its scheduled time bound. The source node increases the transmission range and transmits the PRM again to reserve the routing path towards the CN. By this way, it reserves the path to CN, otherwise the source node directly sends data to the BS.

### 4.5. Data Sensing and Compression

The deployed sensor-based detector gathers data on the sensed data, the position of the explosive material, and an exact location picture. Sensor nodes compress the sensed data to effectively handle the sensor node resource and prolong the network lifetime. The Discrete Cosine Transform (DCT) or Discrete Wavelet Transform (DWT) is used in the image compression techniques to improve the image quality. However, both transformations reveal an increased complexity that requires more time, memory, and processing power. As a result, it has a constraint to apply in WSN since it consumes more energy.The image compression method based on LBT and Golomb + Multiple Quantization (MQ) coders described in [44] has a lower computational cost and memory need, making it perfectly adapted for distributed mobile WSNs. Thus, the compression method described in [44] is used to compress the image in the proposed work.

### 4.6. Significance of the Proposed Methodology

Features of the proposed node deployment and data collection in land mine detection are given as follows:Minimum number of sensor nodes is used to cover a large network region that decreases the complexity and cost of installation.It offers an effective technique for implementing a fully automated land mine detection system that reduces human participation and prevents human life risk.Horizontal search efficiently monitors the affected region of the land mine allowing easy identification of the secure region, while random node deployment does not ensure the entire sensing area.The suggested PRM-ACK based routing avoids collision and congestion. So, it is highly appropriate for the collection of event-driven data.The suggested data routing technique offers a precise location of the buried place of explosive material enhancing the safety of land mine disposal.

## 5. Mathematical Analysis of the Proposed Mechanism

### 5.1. Energy Consumption

Energy consumed by each sensor node depends on the sensing of event, image compression, GPS, exchange of control packets for path reservation, and number of packets processed by each node. Total energy consumed by the sensor node for data transmission $\left(e_{t}^{d}\right)$ is given in Equation (Equation 7).
(7)$$e_{t}^{d}=N_{dp}\times \left(e_{r}\left(k\right)+e_{t}\left(k,d\right)\right)$$
where $N_{dp}$ is the number of data packets processed. Total energy consumed by the sensor node for exchanging the control packets $\left(e_{t}^{c}\right)$ is given in Equation (Equation 8).
(8)$$e_{t}^{c}=\left(N_{PRM}+N_{ACK}\right)\times \left(e_{r}\left(k\right)+e_{t}\left(k,d\right)\right)$$
where $N_{PRM}$ and $N_{ACK}$ are number of PRM and ACK processed by each node. The energy consumption for image processing per bits ($e_{cb}$) [44] is given in Equation (Equation 9).
(9)$$e_{cb}=2\left(e_{pre}+e_{dct}\right)+e_{code}$$
where $e_{pre}$ is the energy dissipation for LBT preprocessing, $e_{dct}$ is the energy dissipation for binary DCT, and $e_{code}$ is the energy consumption for coding.

Noting r as the compression ratio, it’s better for camera sensor to compress the raw data before sending than to directly send the raw data, when $\frac{e_{cb}+e_{t}}{r}<e_{t}$.

Total energy consumed by the total image size ($e_{t}^{ip}$) is given in Equation (Equation 10)
(10)$$e_{t}^{ip}=I_{s}\times e_{cb}$$
where $I_{s}$ is the size of the image. Total energy consumed by each sensor node $\left(e_{t}^{sn}\right)$ is given in Equation (Equation 11)
(11)$$e_{t}^{sn}=e_{t}^{d}+e_{s}+e_{gps}+e_{t}^{ip}+e_{t}^{c}$$
where $e_{s}$ is the energy consumption for sensing and $e_{gps}$ is the energy consumed by the GPS.

### 5.2. End-to-End Delay

The end-to-end delay for data transmission depends on the number of hops, transmission delay $t_{d}\left(t\right)$, data processing time $pr_{d}\left(t\right)$, propagation delay $p_{d}\left(t\right)$, queuing delay $q_{d}\left(t\right)$, and the number of nodes involved between the source and destination.

$l_{d}$ is defined as the link delay of data packet between the nodes, as given in Equation (Equation 12)
(12)$$l_{d}=q_{d}\left(t\right)+t_{d}\left(t\right)+p_{d}\left(t\right)+pr_{d}\left(t\right)$$
M/M/1 queuing model is utilized in the proposed work to find the delay experienced in the queue and it is calculated as
(13)$$q_{d}\left(t\right)=\frac{\frac{\beta k}{b}}{1-\frac{\beta k}{b}}\times \frac{k}{b}$$
where *k* denotes the length of the data packets, *b* is the transmission bit rate, β is the average arrival data rate.

The propagation and transmission delay is determined using Equations (14) and (15)
(14)$$t_{d}\left(t\right)=\frac{k}{b}$$
(15)$$p_{d}\left(t\right)=\frac{t_{r}}{v}$$
where $t_{r}$ is the transmission range, *v* is the propagation speed.

End-to-end delay $\left(e_{d}\right)$ between the source and destination is calculated as
(16)$$e_{d}=n_{hop}\times e_{d}$$
where $n_{hop}$ is the number of hops.

The time complexity for the proposed model is O(n) which depends on the number of intermediate nodes (both routers and MNs).

### 5.3. Control Overhead

MN reserves the path by exchanging PRM and ACK when it detects the target. The maximum path length between the CN and MN ($P_{MN-CN}$) is calculated as
(17)$$P_{MN-CN}=\left(\frac{N_{d}}{2}-1\right)\times t_{r}$$
where $N_{d}$ is the number of mobile nodes or number of strips in the sensing area.

The path length between the CN and BS ($P_{CN-BS}$) is calculated as
(18)$$P_{CN-BS}=N_{ro}\times t_{r}$$
where $N_{ro}$ is the number of router nodes.

The total path length between the CN to BS ($P_{MN-BS}$) is calculated as
(19)$$P_{MN-BS}=P_{MN-CN}+P_{CN-BS}$$

The number of control packets exchanged for the path creation between the MN to CN is $\frac{N_{MN}}{2}$ which is approximately equal to $O(N_{MN})$.

## 6. Simulation Results and Discussion

The proposed protocol uses only a small number of sensor nodes to cover the sensing area and then transmits data to the BS when it detects events in the sensing area. In contrast to the existing protocols, they deploy a large number of sensor nodes to cover the sensing area and then periodically transmit data to the BS. The suggested protocol is examined with EDCGRP, LEACH and MBC using Network Simulator-2 (NS-2) and the performance of the proposed protocols is evaluated with respect to average energy consumption, average end-to-end delay, packet delivery ratio and control overhead. The simulation setup is given in Table 3.

The total targeted sensing area for this experimental setup is 100 × 100 m${}^{2}$ and it is split into strip based on Equation (Equation 2) and has twenty five strips. The total number of mobile nodes required to cover the sensing area is 25. These twenty five nodes deployed at one end of the sensing area and each strip is covered by a single mobile nodes. The energy consumption model for the wireless communication is mentioned in Equations (5) and (6). The typical values of the energy model parameters are $e_{elec}=50\;\mathrm{nJ}/\mathrm{bit}$, $\epsilon_{fs}=10\;{\mathrm{pJ}/\mathrm{bit}/m}^{2}$, $\epsilon_{mp}=0.0013\;{\mathrm{pJ}/\mathrm{bit}/m}^{4}$. The energy consumption of the image compression is considered in this paper as given in [44] which was tested using StrongARM SA1100 processor. The size of the image is 512 × 512 pixels with 8 bits per pixel. The parameters values of the computation energy model are $e_{pre}=15\;\mathrm{nJ}/\mathrm{bit}$, $e_{dct}=20\;\mathrm{nJ}/\mathrm{bit}$ and $e_{code}=90\;\mathrm{nJ}/\mathrm{bit}$ with the bit rate of 0.25 bpp.

The performance of the proposed protocol is evaluated from Figure 6, Figure 7, Figure 8, Figure 9 and Figure 10. The simulation results are obtained for every 60 s over the simulation run time of 360 s. Totally, 25 nodes are deployed horizontally at the sensing zone border (i.e., close to the BS border) and each strip has a single sensor node. And, the transmission range of sensor node is varied from 2 m to 100 m.

It is observed from the Figure 6 that the proposed TRT saves a substantial quantity of battery energy as compared to EDCGRP, MBC and LEACH because it adjusts the transmission power based on distance to neighbors. Also, it utilizes multi-hop data transmission efficiently by choosing the nearest neighbor. The image compression method based on LBT and Golomb + Multiple Quantization coders is utilized in the proposed method which reduces the image size and battery energy consumption. In addition, the routing based on PRM and ACK with hop count ensures the accessibility of links and prevents the exchange of unnecessary control packets. But, when MBC finds the events, it organizes the routing between the source and the BS, so it uses more control packets. Also, MBC met the void issue because of the random deployment of sensor nodes which can’t ensure the connectivity and coverage of the link, resulting in the loss of packets and retransmission of data. LEACH consumes high battery power as it communicates directly with BS for data transmission. When forming a cluster and route, EDCGRP needs additional control packets. It is concluded that the loss of packets, retransmission, and overhead control are directly contributing to energy consumption.

As shown in Figure 7, it is noted that the proposed TRT provides consistent performance of packet delivery ratio in comparison with EDCGRP, MBC and LEACH, because the T-based routing topology ensures the link availability. Also, PRM-ACK based route reservation mechanism prevents the collision and congestion in data routing. In the existing protocols, there is connectivity and link failure issues are occurred owing to random deployment and lack of coordination between the sensor nodes leading to the packet collision and loss. However, in the proposed TRT, it does not create the routing between the source and BS which constructs only up to the coordinator node. Also, the data packet moves through the coordinator to the router nodes and easily routes to BS without collision and congestion. The image compression technique based on LBT and Golomb + Multiple Quantization coders minimizes the image file size which reduces the packet loss and retransmission. As a result, the packet delivery ratio of the proposed TRT is 6.4%, 12.96% and 16.63% higher than the EDCGRP, MBC and LEACH respectively.

From Figure 8, it is obtained that a coordinator-controlled data router ensures a collision-free routing to the BS. Based on the hop count, the proposed T-based routing topology selects the intermediate node which ensures the stable path between the source and coordinator. As a result, it reduces packet loss and makes routing to the BS easier. Also, TRT minimizes the end-to-end delay in packet delivery when compared to the EDCGRP, MBC and LEACH because it achieves collision-free communication between the source and BS by exchanging the PRM-ACK. It is observed from Figure 8 that the average end-to-end delay of the proposed TRT is 10.33% 19.72% and 13.84% lower than EDCGRP, MBC and LEACH respectively.

The energy consumption of the proposed protocol is evaluated with respect to the range of transmission as shown in Figure 9, where the simulation results are obtained over a simulation time of 300 s with 25 sensor nodes. The transmission range of the sensor node in this scenario varies from 20 to 100 m. The network’s availability and coverage are the important parameters to be considered in WSNs to ensure the lossless communication. The impact of the transmission range is analyzed in terms of the energy consumption to improve the energy efficiency and network lifetime. The energy consumption directly attributes to the distance between the transmitter and receiver. It is noted from Figure 9 that the average energy consumption of the proposed TRT is 23.25%, 32.81% and 44.29% lower than the EDCGRP, MBC and LEACH, because it efficiently works with the strip-based node deployment and offers stable accessibility to its neighbors.

It is seen from Figure 10 that the proposed TRT has utilized the minimum number of control packets as compared to EDCGRP, MBC and LEACH. The proposed TRT exchanges the control packets to determine the route during data transmission only. But, EDCGRP, MBC and LEACH protocol utilized excessive control packets to form the cluster and routing. It is seen from Figure 10 that the number of control packets utilized in the proposed TRT is 17.09%, 26.7% and 16.31% lower than the EDCGRP, MBC and LEACH.

In Table 4, the size of the sensing area is varied from 100 m × 100 m to 500 m × 500 m, the number of nodes deployment depends on Equation (Equation 2), here $d_{c}=2$ m. The transmission range of the sensor nodes is varied from 2 m to 100 m for the proposed TRT as well as EDCGRP, MBC, and for LEACH, it depends on the size of the network area. From the simulation result, it is found that the proposed TRT is more energy efficient and scalable as compared to EDCGRP, MBC and LEACH when the size of the network increases. This is because the proposed T-based routing topology efficiently uses its minimum transmission range and prevents long-distance data transmission via direct hop. As a result, the proposed TRT conserves significant amount of battery energy as compared to the existing protocols. Moreover, the proposed TRT protocol effectively uses a minimum number of sensor nodes to cover the entire area of sensing and avoids the issue of void. Also, the proposed TRT has a better performance than EDCGRP, MBC and LEACH in terms of average energy consumption, packet delivery ratio and average end-to-end delay.

## 7. Conclusions

A minimal number of sensor nodes are used by the proposed data routing system to cover a large area, which decreases the cost of node placement and makes it easy to track where land mines have been detonated. It is sensible and safe to monitor and dispose of the explosive material thanks to the GPS and camera-based data gathering, which provides the precise position of the sensed event. The complexity of processing and memory is decreased by using the LBT and Golomb + Multiple Quantization coders based image compression approach. Additionally, it can set up a T-based routing topology between the source and BS to allow event-driven based data gathering. And, it ensures a congestion-free data transmission by reserving the path based on the PRM-ACK mechanism. As a result, the suggested TRT mechanism effectively uses battery power and extends the network lifetime. It delivers better results in terms of average end-to-end delay, average energy consumption, packet delivery ratio and control overhead. Also, it is ideally suited for land mine detection and other applications of a similar nature. In the future, a test bed will be developed to evaluate the performance of proposed protocol for the real time applications. 

## Figures and Tables

**Figure 1 sensors-23-02162-f001:**
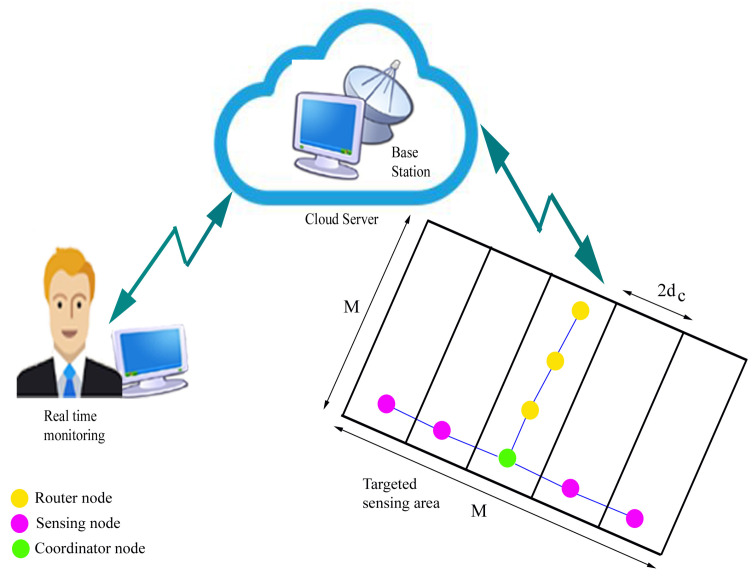
General diagram of proposed model.

**Figure 2 sensors-23-02162-f002:**
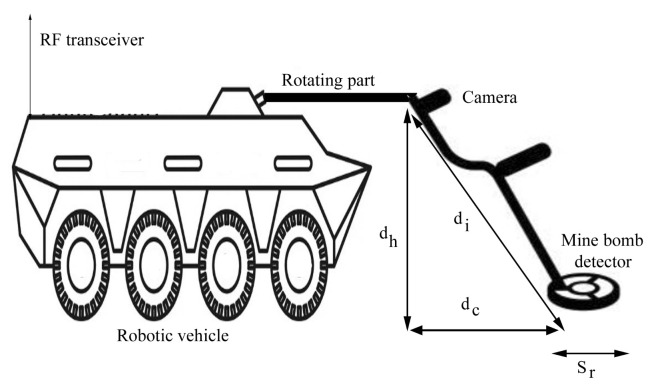
Robotic vehicle with sensor pole.

**Figure 3 sensors-23-02162-f003:**
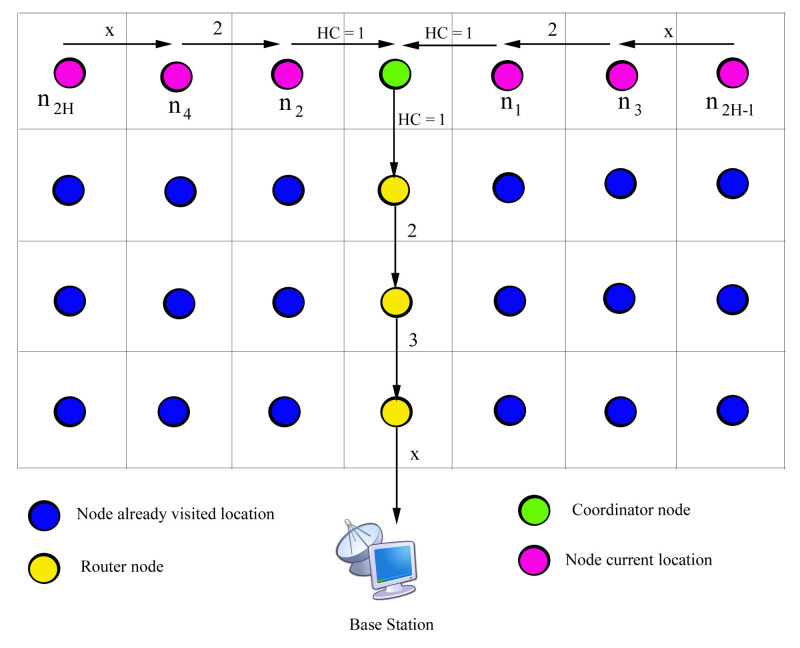
T-based routing topology.

**Figure 4 sensors-23-02162-f004:**
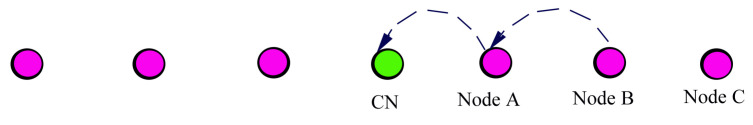
Example showing the usage of wait message.

**Figure 5 sensors-23-02162-f005:**
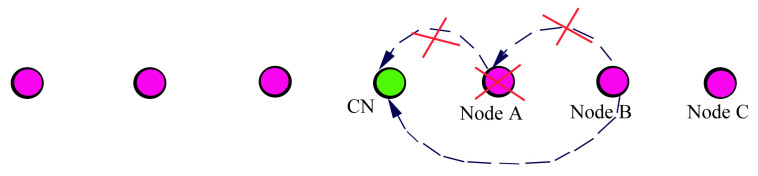
Example showing the fault-tolerance.

**Figure 6 sensors-23-02162-f006:**
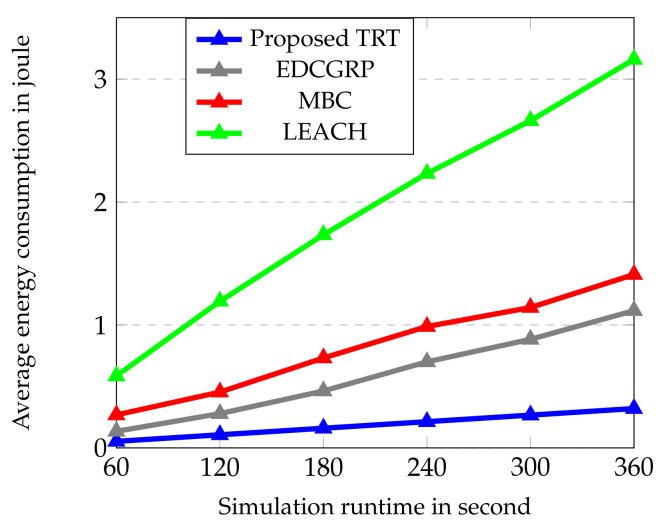
Average energy consumption vs. Simulation runtime.

**Figure 7 sensors-23-02162-f007:**
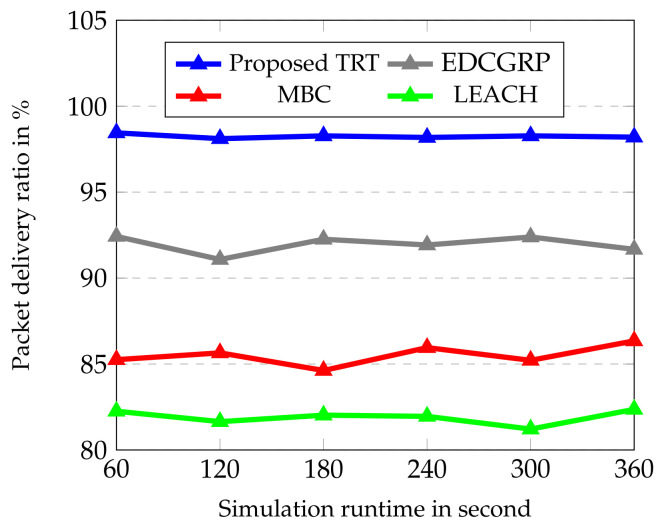
Packet Delivery Ratio vs. Simulation runtime.

**Figure 8 sensors-23-02162-f008:**
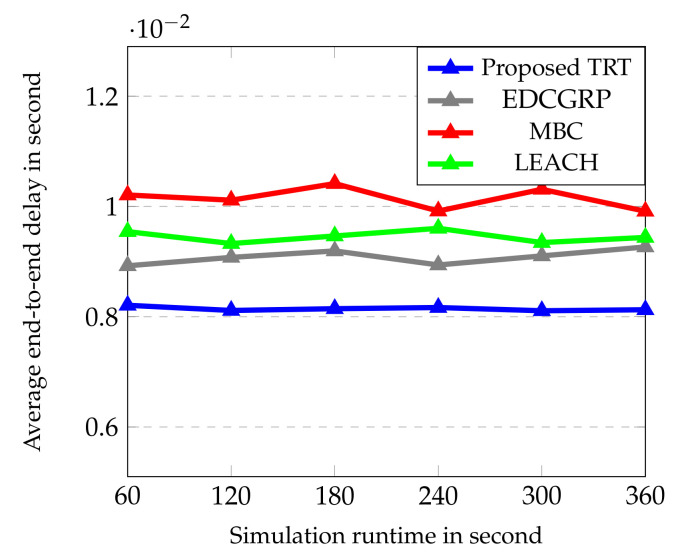
Average end-to-end delay vs. Simulation runtime.

**Figure 9 sensors-23-02162-f009:**
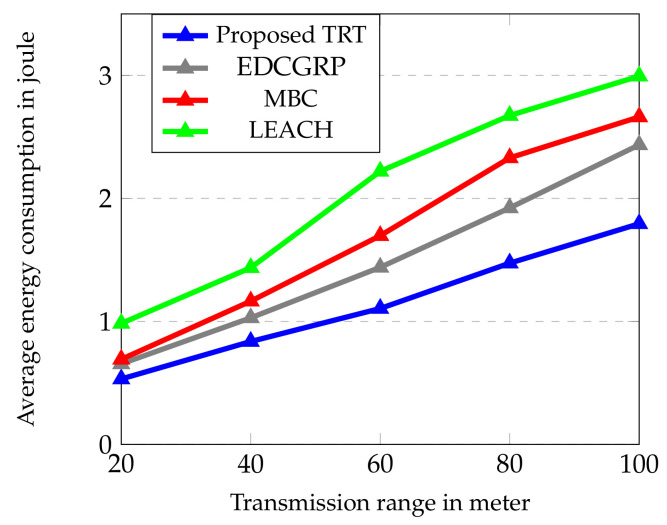
Average energy consumption vs. Transmission range.

**Figure 10 sensors-23-02162-f010:**
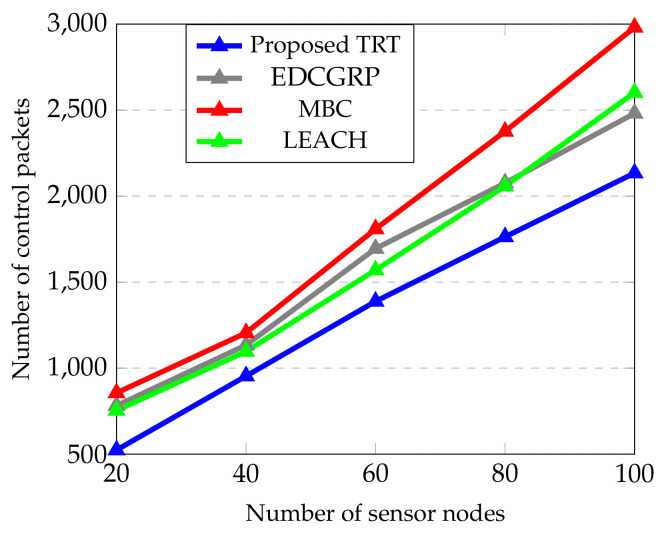
Control overhead vs. Number of sensor nodes.

**Table 1 sensors-23-02162-t001:** Land mine casualties in 2017.

Country	Death
Afghanistan	2300
Syria	1906
Ukraine	429
Iraq	304
Pakistan	291
Nigeria	235
Myanmar	202
Libya	184
Yemen	160

**Table 2 sensors-23-02162-t002:** Comparison of existing methodology with the proposed TRT.

Reference	Mobile Vehicle	Data Communication Mode	Data Collection Methodology	Terrain Suitability	Location Based Real Time Application	Number of Nodes Required	Cost and Maintenance	Scalability	Application	Data Collection Complexity
[35]	UAV	Periodic driven	Clustering and Travelling Salesman Problem	Hilly and flat terrain	No	High	High	limited	Certain condition monitoring example: environment, surveillance	High
[36]	UAV	Periodic driven	zig-zag routing path	Flat terrain	No	High	High	Limited	Smart agriculture	Simple
[37]	USV	Event driven	policy-iteration based path planning	Flat terrain	Yes	Low	Low	High	Target or event detection and tracking (Object detection)	High
[38]	UAV	Periodic driven	bi-level hybridization-based metaheuristic algorithm	Hilly and flat terrain	Yes	High	High	Limited	Forest fire detection	Continuous visit of UAV
[39]	UAV	Periodic driven	Cluster and Metaheuristic Route Planning Algorithm	Hilly and flat terrain	No	High	High	Limited	Certain condition monitoring	Complex
[40]	UAV	Periodic driven	Voronoi diagram based UAV route determination method	Hilly and flat terrain	No	High	High	Limited	Certain condition monitoring	Simple
[41]	USV	Periodic driven	particle swarm optimization (PSO) based path planning algorithm	Flat terrain	No	High	High	Limited	Water monitoring	Complex
[42]	UAV	Periodic driven	K-means clustering strategy	Hilly and flat terrain	No	High	High	Limited	Certain condition monitoring	Simple
[43]	UAV	On-demand or query driven mode	UAV trajectory optimization using the genetic Algorithm	Hilly and flat terrain	Yes	High	High	Limited	Smart farming	Simple
Proposed TRT	USV	Event driven	T—based topology	Flat terrain	Yes	Low	Low	High	Landmine detection	Simple

**Table 3 sensors-23-02162-t003:** Simulation setup.

Simulation Parameters	Values
Targeted sensing network area	100 × 100 m${}^{2}$
Number of sensor nodes	25
Sensor pole coverage radius $\left(d_{c}\right)$	2 m
Base Station location	(50, 0)
Bit rate	50 kbps
Initial energy of sensor nodes	10 joules
Data packet size	512 bytes
Control packet size	25 bytes
Image size	512×512
$E_{elec}$ [26]	50 nJ/bits
$\epsilon_{mp}$ [26]	1.3 fJ/bits/m${}^{4}$
$\epsilon_{fs}$ [26]	10 pJ/bits/m${}^{2}$
$E_{pre}$ [44]	15 nJ/bit
$E_{DCT}$ [44]	20 nJ/bit
$E_{code}$ [44]	90 nJ/bit

**Table 4 sensors-23-02162-t004:** Impact of increase in network size.

Performance Metrics	Protocols	100×100 (m${}^{2}$)	200×200 (m${}^{2}$)	300×300 (m${}^{2}$)	400×400 (m${}^{2}$)	500×500 (m${}^{2}$)
AEC in Joules	TRT	0.30024	0.56247	0.8569	1.2739	1.69354
	EDCGRP [28]	1.1167	1.5023	1.9343	2.4761	2.9902
	MBC [27]	1.41052	1.89421	2.3424	2.8347	3.34556
	LEACH [24]	3.16052	3.94212	4.6547	5.4277	6.5248
PDR in %	TRT	98.1991	97.6245	96.999	96.21	95.67
	EDCGRP [28]	91.672	90.152	88.743	87.032	84.899
	MBC [27]	86.347	85.84	84.29	82.967	82.14
	LEACH [24]	82.362	80.967	80.08	79.65	77.41
AEED in Seconds	TRT	0.00812	0.00832	0.00852	0.00899	0.00924
	EDCGRP [28]	0.00926	0.00939	0.00947	0.00963	0.00985
	MBC [27]	0.00943	0.00964	0.00982	0.01294	0.01372
	LEACH [24]	0.00991	0.01198	0.01401	0.01692	0.01927

## Data Availability

The data presented in this study are available on request from the corresponding author.

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
