# Peer review of "An Energy-Efficient T-Based Routing Topology for Target Tracking in Battery Operated Mobile Wireless Sensor Networks"

_sensors, 2023, doi:10.3390/s23042162_

Round 1
Reviewer 1 Report
1. To highlight the novelty of the proposed model, the authors are advised to add a comparison table exhibiting the differences between the existing works and the proposed work.
2. The manuscript has a good amount of mathematical formulation. However, the manuscript lacks in properly differentiating between the proposed formulation and the existing formulation.
3. In the proposed work, T-based routing topology is used. However, the work lacks in providing solid justification that why this topology is used and not any other.
4. In the proposed work, GPS is used for location tracking. However, it does not give accurate results. The authors are advised to use more advanced location-tracking phenomena.
5. The figures added in the proposed work are not of good quality. Good quality figures should be added.
6. The election of the parameters and the respective values (given in Table 2) should have proper justification for selection.
7. The authors are recommended to add the following sections to the manuscript.
a. Significance/Novelty.
b. Problem Statement.
8. The manuscript has many grammatical errors. Therefore, thorough proofreading is required.

Reviewer 2 Report
This paper proposed a T-based Routing Topology for Target Tracking to obtain information from the land mine-affected areas.I'm afraid this idea is simple that it can't be used in practice. In this paper, the sensor nodes (should be mobile nodes)are used to locate and pinpoint the precise location of buried explosive materials using a camera and the Global Positioning System (GPS).I don't think this architecture is a wise architecture which is proposed in Fig. 1. At present, there are many better methods. For example, the UAV method will be more effective and faster, which has been studied a lot, some of which are as follow:
"GA-DCTSP: An Intelligent Active Data Processing Scheme for UAV-enabled Edge Computing," IEEE Internet of Things Journal, DoI: 10.1109/JIOT.2022.3220840, 2022. DRL-based Trajectory Planning for Unmanned Aerial Vehicles for Data Collection in Dynamic IoT Network. IEEE Transactions on Intelligent Vehicles, DoI: 10.1109/TIV.2022.3213703, 2022. The routing algorithm proposed in this paper is difficult to apply in such applications.Although I made some negative comments, I still gave the author a chance to revise. Because, in most cases, it happened like this, which enlightens me not to give opinions to reject.

Reviewer 3 Report
1. The abstract is not enough to summarize the content of the article.
2. There is no summary in related work, and the common methods based on clustering mechanism have no reference value for this paper.
3. There are many blanks in the article page, which are recommended to be filled.
4. The path length calculation from CN to MN in Section 4.3 is problematic.
5. No details on how to use GPS and cameras to determine the precise location of mines.
6. It is not explained why the network coverage can be improved by adopting the rectangular structure.

Round 2
Reviewer 2 Report
The authors have answered some questions raised by the reviewers. However, their revision has not improved the manuscript, and there are still some major problems to be solved, namely,
1. What is innovation in this paper? How the Network Simulator-2 (NS-2) is extended or revised in a new or innovative way? Clearly indicate the domain-independent innovative advance brought about by the proposed works.
2. The idea of T-based routing topology is not clear. Authors did not present how to manage routing topology, what is T-based, which parameters are shared and revised, what would be optimal number of parameters, etc.
3. What are limitations of T-based routing topology? How the routing topology works in case of different input type? What happens with noise? Can we improve this process?
4. The comparison baselines are too old (all proposed ten years ago). Why not compare the latest SOTA?
Reviewer 3 Report
1. The proportions of the pictures in the article are out of kilter.
2. There is no summary in related work, and the common methods based on clustering mechanism have no reference value for this paper.
3. No details on how to use GPS and cameras to determine the precise location of mines.
4. It is not explained why the network coverage can be improved by adopting the rectangular structure.
5. The location and distribution of events in the sensor area were not detailed in the experimental phase.
